# Auditory Personalization of EMDR Treatment to Relieve Trauma Effects: A Feasibility Study [EMDR+]

**DOI:** 10.3390/brainsci13071050

**Published:** 2023-07-10

**Authors:** Joy Grifoni, Marco Pagani, Giada Persichilli, Massimo Bertoli, Maria Gabriela Bevacqua, Teresa L’Abbate, Ilaria Flamini, Alfredo Brancucci, Luca Cerniglia, Luca Paulon, Franca Tecchio

**Affiliations:** 1International Telematic University Uninettuno, 00186 Rome, Italymassimo.bertoli91@gmail.com (M.B.); teresalabbate95@gmail.com (T.L.);; 2LET’S and LABSS, Institute of Cognitive Sciences and Technologies ISTC, Consiglio Nazionale delle Ricerche CNR, 00185 Rome, Italy; 3Studio Psyche Neuroscienze, 00192 Rome, Italy; 4Psychotherapist Freelance, Perugia, Italy; flamini.usl1@gmail.com; 5Dipartimento di Scienze Motorie, Umane e della Salute, Università di Roma ‘Foro Italico’, 00135 Rome, Italy; 6Luca Paulon, Engineer Freelance, 00159 Rome, Italy

**Keywords:** multisensory sensorimotor stimulation, bilateral alternating stimulation (BAS), psychic trauma (PT), post-traumatic stress disorder (PTSD), audio-visual stimulation, music, personalized therapy, reward

## Abstract

According to the WHO (World Health Organization), Eye Movement Desensitization and Reprocessing (EMDR) is an elective therapy to treat people with post-traumatic stress disorders (PTSD). In line with the personalization of therapeutic strategies, through this pilot study, we assessed in people suffering from the effects of trauma the feasibility, safety, acceptance, and efficacy of EMDR enriched with sound stimulation (by administering neutral sounds synchronized with the guided bilateral alternating stimulation of the gaze) and musical reward (musical listening based on the patients’ predisposition and personal tastes). Feasibility, quantified by the number of patients who completed the treatment, was excellent as this was the case in 12 out of the 12 enrolled people with psychological trauma. Safety and acceptance, assessed by self-compiled questionnaires, were excellent, with an absence of side effects and high satisfaction. Efficacy, quantified by the number of EMDR treatment sessions required to reach the optimal scores on the Subjective Units of Disturbance (SUD) and Validity of Cognition (VOC) scales typical of EMDR protocols, revealed an average duration of 8.5 (SD 1.2) sessions, which is well below the 12 sessions considered a standard EMDR treatment duration. EMDR+ appears to be a relevant personalization of EMDR, particularly in music-sensitive people, consolidating the therapeutic alliance through a multisensory communicative bond for trauma treatment.

## 1. Introduction

Trauma can generate several symptoms, such as intrusive experiences, avoidance attitudes, mood disorders, and alterations in the psychomotor state, which affect both physiological and cognitive functions, inhibiting the normal course of life and learning. The possible altered state following a traumatic event, whether isolated or prolonged, is referred to here as psychic trauma (PT). The ongoing pandemic profoundly affected people’s well-being [1,2], especially impacting caregivers, health workers, and self-employed workers. Among the most affected groups are victims of domestic abuse, due to the segregation linked to the containment of the pandemic [3], and caregivers, with an incidence of between 7% and 37% observed in a population of approximately 70,000 workers [4,5].

The World Health Organization (WHO) approved Eye Movement Desensitization and Reprocessing (EMDR) as an elective therapy against post-traumatic stress disorder (PTSD). The well-established EMDR therapy is based on inducing bilateral rhythmic eye movements (BAS). Its purpose is to defuse the short-circuited pathways that trauma has stabilized, thereby reactivating the ability to rework the traumatic experience through the engagement of large circuits affected and inhibited by trauma [6].

The adaptive capacity of the nervous system emerges from a constant dynamic synchronization interaction between neural networks that manage recursive sensorimotor feedback. The neuroplastic changes that allow the brain to adapt positively to variable environmental demands are based on the correct processing of sensorimotor input signals that allow the calibration of a coherent and synchronous response with respect to perception. This matching depends on a delicate control of modulatory gain related to the hierarchies of the neuronal populations involved [7], and the paradigm in which this balance is achieved concerns the management of sensory-motor balance: a motor action corresponds to sensory feedback, which is useful for calibrating a perceptual adaptation at the level of the plasticity of entire networks [8,9]. In a recent opinion [8] based on a neurophysiological model of the neuronal network organization [9], we suggested that, in clinical practice, the therapeutic alliance can be further enhanced by favoring the choice of deeply evocative and rewarding multisensory input [10,11,12,13].

Scientific literature proves the strong positive impact of sound and musical stimulation on synaptic activity across multiple brain networks, with valuable cognitive and sensorimotor effects [14]. Furthermore, research on music and perception has suggested how the rhythmic-melodic stimulus can support cognitive functions of relearning, acting on brain plasticity at the cognitive-mnemonic level via synchronization and extinction, thanks to the mechanisms of mirroring and reward [15]. Memory deficits related to trauma and the deep encoding of mnesic material can be influenced by the temporal structure implicit in musical stimuli. Practicing and listening to music induces an intense engagement in a pleasant activity, stimulating a positive subjective experience, which is useful for the rapid improvement of the psychophysical state [16].

The present strategy refers to the psycho-acoustic principle based on ‘auditory stream segregation’, ‘fission’, or ‘auditory scene analysis’ [17], in which a multi-layered sound stream is concentrated into a single emergent pattern rich in different perceptions capable of acting on attentional condensation. At the linguistic level, a similar phenomenon (related to speech) is the “cocktail party” effect, which allows the listener to focus on a single voice within a context rich in distractors that implement attentional focalization [18,19,20], revealing for these complex (multi-layer) stimuli an important role in predictive information processing and in retrospective mnesic retrieval [21].

Recent studies on the functioning of sensory-motor neural populations [22] show how certain areas (dorsolateral prefrontal cortex, anterior cingulate gyrus, F5 motor supplementary area, Brodmann primary auditory areas 41–42, Brodmann associative areas 22–44, temporal gyrus, pars opercularis of the inferior frontal gyrus, basal ganglia cerebellum, and caudate nucleus) actively talk to each other, especially in the presence of sound stimuli characterized by temporal periods related to integers (i.e., metric layers) triggering a motor automatism that influences basic physiological arousal and, above all, unconscious psycho-physical coordination [23,24]. Sound-musical stimulation has the potential to synchronize multiple biological oscillators simultaneously. Fluctuations in different time domains tend to synchronize with each other, maintaining proportionate relationships through periods [25,26]:cardiovascular-respiratory [27];

-cardiovascular motor [28];-respiratory motor [29];-metabolic efficiency [30].

While traditional EMDR protocols involve BAS via horizontal eye movements, and the selective use of auditory stimulation appears to be less effective than eye movements [31], it is possible to observe how the addition of synchronous acoustic and tactile stimuli can increase the effectiveness of the protocol [32].

Overall, the above evidence supports that both cortical and limbic musical stimulation offers people with psychic trauma more opportunities to internally build new resilience resources, contributing overall to a better outcome for the therapeutic process. This is the reason why we propose here an enriched EMDR protocol with auditory stimulations via bi-lateralized sounds and musical reward (EMDR+).

To assess the PwPT’s (patients with psychic trauma) musical aptitudes and perceptual abilities in relation to the sound stimuli to be used for bimodal therapy, we selected the Musical Aptitude and Acoustic Skills test (MAAS [33]). Previous experience with the MAAS test showed that listening to music is preferred in moments of leisure; that “cultured” listening genres (classical, jazz, blues) support serenity and relaxation [34]; that music is perceived as an important aid in improving mood while being a distractor during reading [35]; that melody and rhythm are perceived as fundamental elements of musical communication [36]; that actively seeking out pleasant music is important, while passive reception (subliminal listening, like piped music in shops) is experienced as annoying [37]. The MAAS test was initially applied for training and support purposes in the educational field, a context in which its effectiveness in drawing a picture of acoustic capabilities and musical preferences has been repeatedly validated, with ambisexual learners belonging to various developmental ages [38]. The test concerns musical aptitude, probing a person’s natural predispositions and measuring auditory ability and liking for low, long, and medium-low sounds.

The objective of this pilot study is to provide a personalized protocol for treating the effects of psychic trauma by evaluating the feasibility, safety, acceptance, and effectiveness of enriching the EMDR intervention with sound and music (EMDR+).

## 2. Materials and Methods

The design of this pilot study applies the Plan, Do, Check, Act procedure [39], having planned the EMDR+ protocol (Plan), implemented it in a group of people to whom it is dedicated (Do), having evaluated the protocol in terms of feasibility, safety, acceptance, and effectiveness (Check), and having offered it to potential beneficiaries: psychotherapists and, through them, people suffering by trauma via the present publication (Act).

### 2.1. Patient Enrolment

Adult people of both sexes were enrolled (6 females, 6 males, 48.4 ± 15.3 years old) who had developed at least six months of continuous symptoms related to psychic trauma (Person suffering by Psychic Trauma and stress-related disorders, PwPT). Before participating in the study, all participants were provided with detailed information about the procedures and requirements and were asked to provide informed consent. The eligibility criteria for participation, as outlined in the flowchart (Figure 1), were as follows:Inclusion:-Diagnosis of PwPT as assessed by professionals in the healthcare system applying criteria for ‘Disorders specifically associated with stress’ as determined by WHO ICD-11, 6B.

Exclusion:-Symptoms attributable to the physiological effects of a substance or another medical condition (including psychiatric disorders);-Epilepsy or other concomitant neurological or general diagnoses;-Non-corrected major visuo-auditory impairment;-Under treatment with psychotropic drugs;-Comorbidity with psychiatric disorders.

### 2.2. EMDR+ Implementation

The timing of EMDR+ sound enrichment, which is described in the following, is presented in Table 1 in relationship to the settled phases of EMDR standard procedure [40].

#### 2.2.1. Audio-Visual BAS

In the present EMDR+ setting, audio-visual BAS consisted of 24 left—right ear stimulations over 60 s, synchronous with eye movements induced by the therapist who followed the LED on the headset rhythmically oscillating from left to right ears. The sound, equal to the two ears, was a ping at the frequency of 432 Hz adding a variable binaural grounding in progressive decreasing dichotic deviation.

#### 2.2.2. Reward-Song

To identify PwPT’s favorite music pieces, then used as the Reward-Song, at the beginning of the treatment process, all patients were administered with the MAAS test [33]. The test proved to be of great importance in understanding the musical preferences of enrolled PwPT so that each was assigned a pleasant listening experience in the involved phases.

The MAAS test was administered online (web compilation via a private link) to allow each PwPT to fill in the answers with the utmost confidentiality and in a situation of domestic tranquility ideal for concentration, which is of fundamental importance, especially for the listening part. By filling in the MAAS test, patients selected a pleasant, familiar, and reassuring piece of music, thought to be an additional means of encouragement, consolation, and protection. The selection was provided among pieces of music with a slow and steady rhythmic pattern, rich in grave sounds, with a preponderant harmonic key of major mode. More frequently, instrumental rather than sung pieces were preferred by most users.

#### 2.2.3. Key-Song

The Key-Song is a piece of music (chosen by the patient when filling in the MAAS test) that can help him/her to recall the traumatic memory as it is emotionally or symbolically linked to the contents to be recalled for the re-elaboration of the painful memory [41].

Patients had full control over access to the stimulus: the open headphones used allowed optimal listening to the therapist’s voice and environmental noises (avoiding the typical sense of estrangement and isolation of listening with insulating headphones) and were equipped with a volume controller entrusted directly to the user, so that he/she could decide autonomously at any time whether to enjoy the music.

### 2.3. Feasibility Assessment

We assessed the EMDR+ feasibility by counting the patients’ dropouts. No dropout was registered.

### 2.4. Safety Assessment

The safety of EMDR+ was assessed by evaluating side effects, investigated by a questionnaire collected at the end of the therapeutic pathway.

### 2.5. PwPT Acceptance Assessment

At the beginning and at the end of therapy, all patients were administered a Cognitive Behavioral Assessment (CBA [42]) to assess the overall satisfaction. The ranges of the score of the five domains’ subscales are: anxiety (−4, 52), well-being (0, 60), perception of positive change (−4, 40), depression (−8, 68), and psychological distress (0, 84).

### 2.6. Efficacy Assessment

We considered the EMDR+ treatment complete when SUD and VOC reached the ‘healthy’ value. That is, when at the end of a session at least one out of SUD reached the 0 score [SUD ranges from worst 10 value to optimal 0], and/or VOC reached the 7 score [VOC ranges from worst 0 to optimal 7]. The EMDR+ efficacy was quantified by the total number of sessions from the beginning of the treatment to such a criterion.

To assess the dynamics of the psychic state throughout the treatment, we included both SUD and VOC in a single index that was defined as the difference VOC-SUD (varying from –10 to 7) so that a state amelioration corresponds to an increase in the VOC-SUD index. We followed the state evolution via the change between the end and the beginning of the session when each treatment phase occurred.

Finally, the effect size (ES) was estimated by the coefficient Cohen’s d = (*M*_2_ − *M*_1_)⁄*σ* = √(*σ*_1_^2^ + *σ*_2_^2^)⁄2 on both SUD and VOC scales. We considered the SUD and VOC scores at the beginning of the treatment (t0 of the first session) and at the end of it (t1 of the last session) across the enrolled population.

### 2.7. Data Analysis

For the EMDR+ feasibility, safety, PwPT acceptance, and efficacy, we applied the above criteria in the enrolled population.

To describe the dynamics of the therapeutic process, we calculated the change between the VOC-SUD (t1) and the VOC-SUD (t0) in the session when each phase was concluded. The single subject values of these variations were subjected to an ANOVA model with the within-subject factor Phase (Anamnesis, Preparation,…, Revaluation) to examine whether there was a significant Phase effect indicating that the clinical state improved more in specific phases.

Finally, we compared the Cohen’s d coefficient with the known thresholds. In fact, a Cohen’s d ≤ 0.2 indicates a small ES, 0.2 < d < 0.5 a medium ES, 0.5 < d< 0.8 a large ES [43], and a further classification evaluates as very large the effects with corresponding Cohen’s d above 1.2 and huge above 2 [44].

Significance *p* values < 0.05 were considered significant.

Statistical analysis was performed by SPSS v28.0.

## 3. Results

Eleven of the 12 PwPTs enrolled (Table 2) were persons suffering from the consequences of COVID-19, being caregivers such as family members or health system personnel. The person with trauma symptoms that arose ten years ago was suffering from a family bereavement.

### 3.1. Feasibility

All of the 12 PwPT successfully concluded the therapeutic pathway, revealing a perfect feasibility of the proposed therapeutic protocol.

### 3.2. Safety

All patients reported no side effects at the end of the therapeutic pathway. None, in fact, perceived any dizziness, nausea, headache, pain, insomnia, or general discomfort.

### 3.3. Patient Acceptance

Patients’ acceptance of the EMDR+ treatment, assessed via the five CBA domains, revealed a clear satisfaction in both positive (well-being and perceived change) and negative (anxiety, depression, and distress) perception counterparts (Table 3).

### 3.4. Efficacy

The average duration was 8.5 (Standard Deviation 1.2) sessions (Figure 2).

The evaluation of the state dynamics (Figure 3), assessed via the VOC-SUD changes before and after each phase and submitted to the devoted ANOVA, indicated a strong Phase effect [F(7, 77) = 25.9, *p* < 0.001]. The variations were significant between desensitization and assessment (F(1, 11) = 5.9, *p* < 0.034), installation and desensitization (F(1, 11) = 24.8, *p* < 0.001), installation and body scan (F(1, 11) = 48.0, *p* < 0.001), body scan and closure (F(1, 11) = 48.0, *p* < 0.001), and between closure and revaluation (F(1, 11) = 41.3, *p* < 0.001).

Evaluating the effect size, we observed:SUD Cohen’s d = (4.68 − 7.31)⁄ 0.43 = −6.1
VOC Cohen’s d = (4.29 − 2.84)⁄ 0.67 = 2.0.

## 4. Discussion

The key advancement of our work is the provision of an enriched personalized EMDR protocol, which showed efficacious repercussions in the resolution of symptoms and in the perception of change and well-being via the multimodal EMDR+ stimulation. We observed signs of EMDR+ being an effective therapeutic tool in the treatment process, with short treatment duration and both SUD and VOC changes revealing effect sizes well above the threshold indicated for large efficacies. As a response to treatment, we focused the description on SUD and VOC as part of the EMDR protocol, in this way providing a valuable tool independent of the diagnostic procedures. We note that, while traditionally the SUD and VOC scales are only administered in the desensitization and reprocessing phases, in the presented protocol they were collected at the beginning and end of each session as a tool to monitor the patient’s self-perception. The therapist noted that this was important feedback, enriching the content that emerged during the typically dialogue-based progress monitoring.

### 4.1. Music Expected to Support Recovery from Trauma

The enrichment of the EMDR treatment is based on the ability of multisensory BAS to support the re-emergence of a neurobiological condition of reordering, promoting the integration of traumatic memories at the cortical level, and also thanks to a process of normalization of activation levels concerning the amygdalic-hippocampal structures [45]. EMDR stands in contrast to the functioning that underlies learning by operant conditioning. It is possible to desensitize from a negative prediction (due to trauma) through re-exposure to it (re-enactment) in a setting of absolute safety in which the patient can feel protected and reinforced by positive encouragement. This makes it fundamentally important to offer the PwPT a positive stimulus that functions as a pattern for the desensitization necessary for therapeutic reprocessing; for example, it was already observed that there was a valuable reward consisting in listening to pleasant music [46]. The present protocol further suggests the potential of choosing an appropriate musical reward to provide to the persons suffering from the effects of trauma as therapeutic support and encouragement, which involves a high degree of personalization of the reward as a decisive ingredient for a successful outcome.

### 4.2. Efficacy of Music in Desensitization and Body Scan Phases

We observed a specific effect of music in enhancing the improvement of the psychic state, particularly during the assessment, desensitization, and installation phases. This effect of musical reinforcement aligns with the concept of “reward.” Previous observations have indicated that positive reinforcement through music can positively influence patients’ emotional attitudes, leading to increased gratification and attentional-mnemonic arousal, which facilitates cognitive processing in neural networks [27,47,48]. The progressive desensitization treatment aims to address the consequences of trauma and stress-related disorders and consolidate the dysfunctional implicit memory that contributes to negativity (intrusive thoughts, impact on social relations, self-perception, self-esteem, etc.) and physical symptoms (headaches, chest and abdominal pain, dizziness, gastrointestinal problems, insomnia, tremors, sexual dysfunction, etc.) [49,50].

By incorporating moments of reassurance through gratifying musical listening, we introduced a valuable “cognitive interweave” into the therapeutic process. This integrative cognitive intervention helps make traumatic memories less emotionally disturbing, allowing for their reprocessing. The relatively short treatment duration aligns with music’s ability to influence long-term memory and rapidly reintroduce missing information necessary for a different understanding of past events that were previously inaccessible to recollection. This approach strengthens personal resilience resources, particularly during challenging stages of the therapeutic process such as desensitization and reprocessing, which exhibited more pronounced improvements in the psychic state.

Listening to a specific “Key-Song” that aids in constructing a mental “safe place” proved to be a successful strategy in preventing traumatic relapse. Based on the scores obtained from the VOC and SUD scales, musical listening, serving as a reassuring framework, effectively supported patients in addressing the cognitive, emotional, and sensory components of trauma. This approach yielded notable positive effects on the physical sensations associated with trauma. The proposed holistic approach to traumatic memory proved to be a valuable support for therapeutic intervention [51,52].

### 4.3. Music-Enriched Reward

Reward, the key catalyst for memory consolidation, acts also on maladaptive symptoms in both cognitive and somatization domains [32,53]. Today’s experimental research carefully examines the neural basis of non-declarative memory (such as habit formation, classical conditioning, and fear conditioning) by observing how functional implicit learning via reward influences both cognition and the patient’s physical response. Long-term memory modification operated by trauma affects synaptic transmission within those networks, which also encode compulsions, addiction, anxiety, and phobias. Compulsions and stereotypies involve automated implicit motor skills that are dysfunctionally implemented (e.g., cortico-basal ganglia cyclic aberration [49]). In such a case, reward-based associative learning can induce positive long-term changes in circuitry activations in those areas of the brain that serve basic biological needs, with significant positive correlations at the physiological level. If most somatizations at the physical level are uncontrolled, repetitive defensive reactions (secondary to abnormal fear conditioning), then enrichment with multimodal signals supporting implicit associative learning can be an excellent tool for reactivating adaptive plasticity mechanisms.

### 4.4. Music-Enriched Body Scan

Sound-musical stimulation, performed during the body scan, has been shown to be a powerful source of pleasure and reward for most people [15,54,55]. In line with the central idea of reinforcement learning, the pleasurable and rewarding states promoted by music may function as a catalyzing force of motor behavior. Mediated by automatic brainstem responses triggered very quickly in brain processing by auditory stimulation [56], sounds have effects on attention and physiological arousal: listeners will be attracted to music that induces an ‘optimal’ level of physiological arousal. Theories of musical cognition [57] and musical expression [58] also rely heavily on reward prediction: the ability to anticipate and predict musical events has been recognized as a powerful source of psychophysical pleasure. Furthermore, prediction concerning auditory-motor synchronization of bimodal (visual-auditory) stimulation profoundly impacts associative learning [57,58,59,60].

Acoustic stimulation also has been useful in implementing relaxation in the self-perceptive phases (especially during the body scan process in the sixth phase of the treatment). Music, in fact, is deeply linked to both the sensory tactile-acoustic perceptual system and the motor system, for which it is a powerful trigger of entrainment [61], and can act as a bridge for the dialogue between complementary networks fundamental to self-perception, such as the sensory and motor systems.

### 4.5. Multisensory Treatment

From the pilot trial results, we can speculate that the EMDR+ therapy has been able to interact with those regions of the nervous system that are poorly accessible only to linguistic processing combining verbal stimulation with other types of sense-motor input, such as visual, tactile, and sound. Among these, the combination of visual-auditory input seemed to affect significant improvements, both in the desensitization and reprocessing phases and in the process of perceptual reintegration related to body scanning. This study aimed to highlight, in an experimental way, the significant contribution of sound and musical stimulation within the process of psychophysical reorganization for the well-being of the person in the treatment of psychological disorders. The most likely hypothesis that justifies this evidence is related to the high degree of connection between input and neural networks: similar dysfunctional memories tend to associate in common cognitive networks, so access to memory can also activate those semantically, emotionally, and (above all) sensorially. Having multiple channels of sensory stimulation available promotes access to larger and more complex neural networks. The entire EMDR+ therapeutic cycle (based on tactile and visuo-auditory stimulation) was based on multifactorial procedures that determined an excitation of different associative networks, allowing various forms of access to the dysfunctional areas of memory. This allowed for a better passage of information between neural networks of different specializations and the stored information was recovered more easily according to a higher activation gradient.

Increasing sound and musical stimulations supported a positive treatment outcome. According to the AIP model [62], when a target memory is processed all memories related to similar events are also reactivated through associative memory networks. In this way, the new positive cognitions and emotions can extend to all events clustered in common neural networks.

In the reprocessing of the memory, all of the different therapeutic stimulations were fundamental. The focusing of attention on the event (together with the physical sensations, emotions, and bodily sensations), ontherapist’s fingers, and sounds proposed through the headphones jointly contributed to arousing associations and to restarting the learning process of the event, stored in a new form, allowing the memory to be reprocessed in the emotional memory in an adaptive and no longer dysfunctional manner.

EMDR’s focus on the various cognitive, physiological, and somatic aspects of the processing of traumatic memories leads to the extension of benefits to all these components (cognitive, emotional, and physiological) and toward a progressive, more adaptive change to stress [63].

Constant work on body sensations, aided by sound-music stimulation (especially in the body scan and reward phases) helps the patient gain mastery and management of what is happening at the somatic level. Thus, it appears to be directly related to the decrease in scores on the discomfort scale, which shows a decrease in psychological distress related to the psycho-physical symptoms present at the beginning of therapy, especially in the desensitization and installation phases while listening to music pieces.

Within the EMDR’s working memory hypothesis [64], the more the bilateral stimulation is rich in stimuli, the more effective it is in unblocking the brain’s information processing centers, enriching the connection between the aversive information and the response to a current non-traumatic stimulus. The resulting relaxation response induces physiological responses that, linked to stored information about previous aversive experiences, generate new positive information that is functionally reintegrated [32].

### 4.6. EMDR+ Treatment for COVID-19 Related PTSD

A feature of the present pilot study is the enrolment of persons with psychic trauma. Now that the EMDR+ protocol is available, it would be very interesting to offer it to populations with PTSD in all therapeutic communities of psychiatrists and psychologists. Traumatic experiences related to this pandemic era can frequently lead to PTSD, providing the paradigm of a dramatic alteration to the adaptive physiological nature of the sensory-motor system. Generally, the traumatic experience associated with the dysregulation of neural, neurochemical, and neurobiological mechanisms [65] constitutes a shift from a dynamically adaptive mechanism to a maladaptive one underlying the psychopathological condition. The core of the therapeutic efficacy of EMDR+ therapy is the rebalancing through sensorimotor interaction between therapist and patient, mainly related to bilateral multisensory stimulation that can restore the alteration induced by traumatic experiences on brain activity. A traumatic event, whether single or repeated, implies exposure to a threat concerning the physical integrity of the person, associated with a psychic correlate of fear, helplessness, and vulnerability. PTSD is one of the possible consequences of such an event [66] and occurs with different levels of prevalence related to time and level of exposure to trauma [67].

The present approach can be further integrated by a vast experience in treating chronic fatigue [68,69,70], as this symptom, already impacting people during seasonal influenza [71], has been found to be a main consequence of COVID-19 [72].

We observed a quick recovery from the single trauma induced by the pandemic, corresponding to a short duration of the symptoms of the enrolled people in all cases but one. He was an elderly person who also responded well to the EMDR+ proposed intervention.

### 4.7. Future Directions of Treatment Personalization

In future developments, we will quantitatively consider the acceptance by both the PwPT and the therapist. In the body scan phase, we will direct each patient in self-assessment of their mental and physical well-being by guiding her/his attention to diaphragmatic breathing and body relaxation to detect any contracture, pain, or general discomfort. As the information processing progressed, patients experienced more adaptive attitudes, thoughts, and feelings, which also positively affected their sense of self-esteem and self-efficacy.

Here we have assessed the acceptance of EMDR+ by persons with psychic trauma. In the future we will also consider the acceptance of the therapist, knowing that the bilateral therapeutic alliance benefits profoundly from the gratification of both persons involved. Moreover, the focus on therapist and patient satisfaction is a key to our therapeutic proposal. It allows us to assess the involvement of both in pursuing the main goal of all human actions, i.e., to focus individual resources on the enjoyment of the present condition by evaluating all available reserves and to mutually support each other in achieving this goal, with the clear asymmetry that occurs in a psychotherapeutic process.

The proposed treatment is among the possible therapeutic strategies of personalized medicine [67,68,69,73] for the treatment of trauma starting from the detection of individual perception parameters. It is therefore most effective as it aims to intervene in a highly personalized target by establishing progressive and highly customized goals.

The present protocol is especially suitable for specific personalization, considering the personality trait in selecting the music contents. This is according to the orientation of the standardized S.T.O.M.P test [74,75], traditionally used to assess preferences in musical genres. In its original version, the test identifies five dimensions of musical preference (Mellow, Unpretentious, Sophisticated, Intense, Contemporary) in relation to fundamental personality factors (‘The Structure and Personality Correlates of Music Preferences’ [76]). Both the two parts of which the MAAS test is made up aim to identify, for each profile, the dominant personality traits in relation to the B5F [big five factors] scheme [77], i.e.,:-Degree of extroversion (dynamism-dominance);-Degree of agreeableness (empathy-cooperativeness);-Capacity for conscientiousness (conscientiousness-perseverance);-Predisposition to neuroticism (stability-control);-Open-mindedness potential (culture-experience).

The relationship between musical listening aptitude and personality domains has been investigated several times in the past through numerous tests including the Meier-Seashore Art Judgement Test [78], I.P.A.T. Music Preference Test of Personality [79], Computerized Adaptive Testing of Musical Aptitude [80], and T.I.P.I Ten Item Personality Inventory [81].

## 5. Conclusions

In pursuit of our goal of providing a personalized protocol for treating the effects of psychic trauma, we believe that enhancing the EMDR protocol through multi-sensory stimulation, in particular through the use of music as a reward and tool to evoke the target memory by recalling images, emotions, and physical sensations, provides a very capable means of alleviating the effects of trauma in trauma- and stress-related disorders.

The proposed treatment is among the possible therapeutic strategies of personalized medicine for the treatment of trauma starting from the detection of individual perception parameters. It is therefore most effective as it aims to intervene in a highly personalized target by establishing progressive and highly customized goals.

## Figures and Tables

**Figure 1 brainsci-13-01050-f001:**
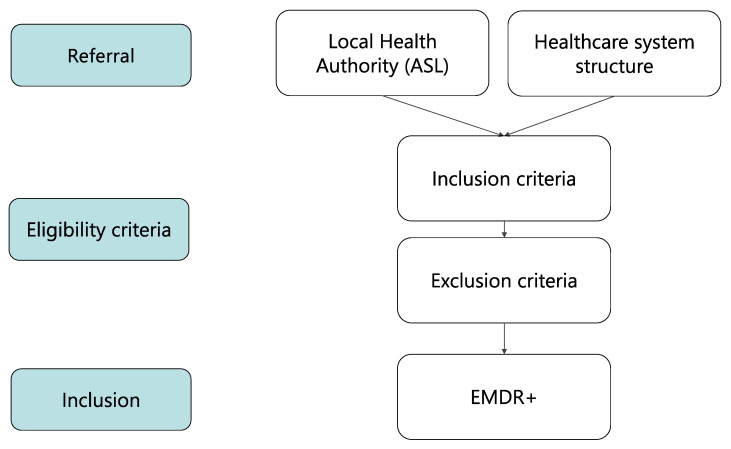
enrollement flowchart.

**Figure 2 brainsci-13-01050-f002:**
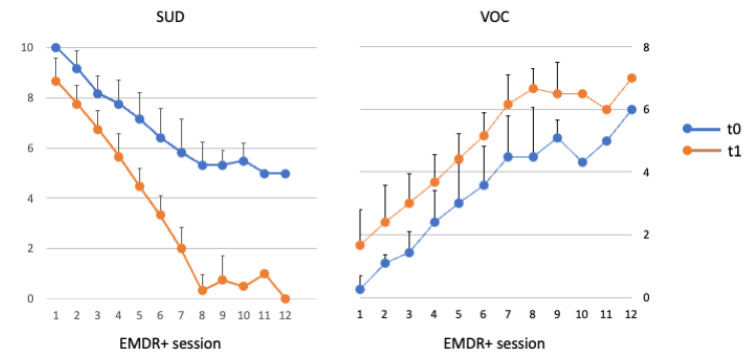
Mean and standard deviation of Subjective Units of Disturbance (SUD) and the Validity of Cognition (VOC) at the beginning (t0) and end (t1) of each session. Note that all 12 PwPT received a minimum of eight sessions, only one PwPT executed nine sessions, one PwPT executed 10 sessions, one PwPT executed 12 sessions (this is the reason why sessions 11 and 12 have standard deviation equal to 0). In addition, note that SUD was initially 0 in all 12 patients (resulting in standard deviation equal to 0).

**Figure 3 brainsci-13-01050-f003:**
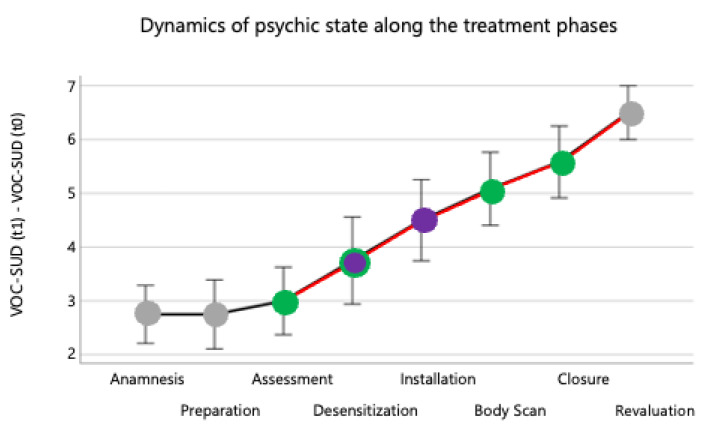
Averages and standard deviations between the 12 PwPTs of the change in VOC-SUD between the beginning and end of each treatment phase. In black are the connectors for the variations that do not differ, in red the significant ones. In grey are the phases without adjunctive sounds, in purple the phases with audio-visual BAS, and in green those when the PwPT listened to music (Reward- and Key-Song).

**Table 1 brainsci-13-01050-t001:** EMDR procedure’s phases and EMDR+ enrichment.

Phase	EMDR	EMDR+
1. ANAMNESIS	Therapist develops the therapeutic plan depending on the patient’s history and functioning, considering health, work, and social status, and identifying the PTSD triggers. The therapist scores the CBA.	After CBA scoring, through the MAAS test the patient identifies her/his favorite music piece (Reward-Song) and evocative music piece (Key-Song).
2. PREPARATION	Therapist explains the EMDR treatment, obtains informed consent, and teaches relaxation techniques by encouraging the patient to develop new coping strategies (deep respiration, muscular relaxation).	
3. ASSESSMENT	The therapist leads the patient in evocating her/his target memory by recalling images, emotions, physical sensations, and associated thoughts identifying her/his positive cognition identifying and assessing her/his secure place. SUD and VOC collection.	The patient listens to Reward-Song to increase concentration on safe place.
4. DESENSITIZATION	Visual BAS, while the patient focuses on the traumatic event until SUD is 0 and/or VOC is 7 new thoughts, sensations, images, and feelings, may emerge.	Audio-visual BAS instead of visual BAS. Patient listens to Key-Song. The patient listens to Reward-Song (secure place) to contain spill over.
5. INSTALLATION	During visual BAS the patient reinforces positive cognition during trauma revocation until it is over.	Audio-visual BAS instead of visual BAS. The patient listens to Key-Song.
6. BODY SCAN	Patient focuses on both the PTSD triggers and the positive cognition while scanning the body from head to toe. Any lingering disturbance from the body is reprocessed.	The patient listens to Reward-Song
7. CLOSURE	Patients are assisted to return to a state of calm, considering if the reprocessing is complete. Reprocessing is finalized when SUD = 0 and VOC = 7. The therapist scores the CBA.	The patient listens to Reward-Song.
8. REVALUATION	Patient and therapist discuss recently processed memories to ensure that distress is still low, and that positive cognition is still strong. Future targets and directions for continued treatment are determined.	

EMDR phase (first column), with the therapeutic process typical of EMDR strategy (second column), and the auditory/musical enrichment that was integrated in addition to the typical EMDR action (third column).

**Table 2 brainsci-13-01050-t002:** People with psychic trauma treated by EMDR+.

PwPT	Sex	Age (years)	Symptoms’ Duration (months)	Sessions (#)
Ptsd1	F	34	6	8
Ptsd2	F	49	6	9
Ptsd3	M	64	6	8
Ptsd4	M	61	6	8
Ptsd5	F	56	8	8
Ptsd6	F	27	7	10
Pysd7	M	58	6	8
Ptsd8	M	71	120	12
Ptsd9	M	47	15	8
Ptsd10	F	22	6	8
Ptsd11	M	38	6	8
Ptsd12	F	54	6	8
Mean	Six Female Six Male	48.4	8.3	8.6
sd		15.3	3.8	1.2
(min, max)		(22, 75)	(6, 120)	(8, 12)

Enrolled population’s sex, age, trauma effects’ origin, symptoms’ duration before the enrolment, and EMDR+ treatment duration in terms of the number of treatment sessions.

**Table 3 brainsci-13-01050-t003:** Cognitive Behavioral Assessment (CBA) quantifying PwPT acceptance.

Domain	Baseline	After EMDR+	*p*
WELL-BEING	11.2 ± 1.5	36.2 ± 4.0	<0.001
POSITIVE CHANGE	19.9 ± 4.0	34.9 ± 3.8	<0.001
ANXIETY	38.3 ± 4.0	8.3 ± 1.6	<0.001
DEPRESSION	53.3 ± 1.9	19.8 ± 1.7	<0.001
DISEASE	46.7 ± 2.5	25.8 ± 2.3	<0.001

Mean and standard deviation of the scores of the specific domains of the CBA test across the 12 people with psychic trauma (PwPT), at the beginning (Baseline) and after the last treatment session (After EMDR+). Paired sample *t*-test significance (*p*).

## Data Availability

The data that support the findings of this study are available from the corresponding author upon reasonable request.

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
