# Peer review of "Auditory Personalization of EMDR Treatment to Relieve Trauma Effects: A Feasibility Study [EMDR+]"

_brainsci, 2023, doi:10.3390/brainsci13071050_

Round 1
Reviewer 1 Report
Dear authors, it is a pleasure to review your manuscript. I would like to make a few comments on it:
-Who confirmed the presence of psychic trauma symptomatology in the patients?
-The signature of the informed consent form is not an inclusion criterion, but a requirement to take part in the study. The authors do not establish inclusion criteria, so there is an important methodological error.
-A flow chart for patient selection should be included.
-Conclusions should clearly respond to the objectives of the study, and are very brief.
-The objective of the study should be clearly stated at the end of the introductory section.
-The study design is not explicitly stated.
Author Response
Dear authors, it is a pleasure to review your manuscript. I would like to make a few comments on it:
- Dear Sir/Madam, thank you very much for your insights into our work, which has benefited greatly from our intervention under your guidance.
- Who confirmed the presence of psychic trauma symptomatology in the patients?
- The signature of the informed consent form is not an inclusion criterion, but a requirement to take part in the study. The authors do not establish inclusion criteria, so there is an important methodological error.
- We removed the error and clarified the process of assessing the psychic trauma symptomatology (pg. 3).
- A flow chart for patient selection should be included.
- We thank the Reviewer for the useful suggestion, and we introduced the flowchart (pg. 4).
- Conclusions should clearly respond to the objectives of the study, and are very brief.
- In the conclusions, we have expressed more clearly how the work responds to the main objective (pg. 13).
- The objective of the study should be clearly stated at the end of the introductory section.
- We expressed more clearly in the introduction the objective of our work (pg. 3).
- The study design is not explicitly stated.
- We stated explicitly the study design at the beginning of the Methods’ section (pg. 3).
Reviewer 2 Report
I think that the subject of the study or the evaluation of musical aptitudes and perceptual abilities, I think it is of great interest
The work is interesting, but I think that the summary should include the context of the study and the characteristics or profile of the participants.
In the method section, I think you should indicate the number of male and female subjects, as well as the ages of the participants.
The procedure is fine
The analyzes carried out are fine.
The presentation of the results is fine.
discussion is fine
Undoubtedly, the discussion section is very enriching, therefore I would ask the authors for more forceful conclusions and indicate that this study supports effective personalization to heal the effects of trauma. More studies like this are needed
Author Response
I think that the subject of the study on the evaluation of musical aptitudes and perceptual abilities, I think it is of great interest
The work is interesting, but I think that the summary should include the context of the study and the characteristics or profile of the participants.
- We are very pleased with the reviewer's appreciation of our work. We included in the summary the context and the participants’ characteristics.
In the method section, I think you should indicate the number of male and female subjects, as well as the ages of the participants.
- We moved in the method section the information about the enrolled population (pg. 3).
The procedure is fine
The analyzes carried out are fine.
The presentation of the results is fine.
- We thank the Reviewer again for her/his kind cooperation.
Round 2
Reviewer 1 Report
Dear authors, after making the suggested changes, the text meets the requirements for acceptance.